# Design and Validation of the INCUE Questionnaire: Assessment of Primary Healthcare Nurses’ Basic Training Needs in Palliative Care

**DOI:** 10.3390/ijerph182010995

**Published:** 2021-10-19

**Authors:** Isidro García-Salvador, Encarna Chisbert-Alapont, Amparo Antonaya Campos, Jorge Casaña Mohedo, Clara Hurtado Navarro, Silvia Fernández Peris, José Bonías López, Maria Luisa De la Rica Escuín

**Affiliations:** 1Nurse Oncology Service, Valencia Health Department, Doctor Peset, 46017 Valencia, Spain; isidro.gs@hotmail.com; 2Foundation for the Promotion of Health and Biomedical Research in the Valencian Region (FISABIO), 46020 Valencia, Spain; camposampa@gmail.com (A.A.C.); jcasamo@gmail.com (J.C.M.); klaracha50@gmail.com (C.H.N.); silviafp9@gmail.com (S.F.P.); boniasjose@gmail.com (J.B.L.); marisadlrscn@hotmail.com (M.L.D.l.R.E.); 3Research Group INCUE, Valencia Health Department, Doctor Peset, 46017 Valencia, Spain; 4Nurse Oncology Service, Valencia Health Department La Fe, 46026 Valencia, Spain; 5Primary Care Nursing Director, Valencia Health Department, Doctor Peset, 46017 Valencia, Spain; 6Nursing Department, Faculty of Medicine and Health Sciences, Universidad Católica San Vicente Mártir, 46001 Valencia, Spain; 7Nurse Training Service, Valencia Health Department, Doctor Peset, 46017 Valencia, Spain; 8Carena Association of Psycho-Oncology, Valencia Health Department, Doctor Peset, 46017 Valencia, Spain; 9Nurse Primary Care Center of San Marcelino, Valencia Health Department, Doctor Peset, 46017 Valencia, Spain; 10Nurse Research Group on Care in End-of-Life Processes, Institute for Health Research Aragón, 50009 Zaragoza, Spain

**Keywords:** questionnaire development, palliative care, nursing, primary healthcare, educational needs’ assessment

## Abstract

Many instruments have been created to measure knowledge and attitudes in palliative care. However, not only is it important to acquire knowledge, but also that this knowledge should reach patients and their relatives through application in clinical practice. This study aimed to develop and psychometrically test the INCUE questionnaire (Investigación Cuidados Enfermeros/Investigation into Nurses’ Care Understanding of End-of-Life) to assess the basic training needs of primary or home healthcare nurses in palliative care. A questionnaire was developed based on the classical theory of tests and factor analysis models. Initially, 18 experts developed 67 items in two blocks and determined content validity by two rounds of expert panels. Exploratory factor analysis and reliability testing were conducted with a non-probabilistic sample of 370 nurses. Some items were observed to have very low homogeneity indices or presented convergence problems and were eliminated. Questionnaire reliability was 0.700 in the theoretical block (KR20 Index) and 0.941 in the practical block (Cronbach’s alpha). The model converges and shows an adequate fit, specifically CFI = 0.977, TLI = 0.977 and RMSEA = 0.05. The correlation between the two factors in the model is ρ = 0.63. The questionnaire objectively evaluates primary or home healthcare nurses’ knowledge of palliative care and its practical application, thereby facilitating more efficient training plans.

## 1. Introduction

Palliative care (PC) improves the quality of life of patients facing life-threatening illnesses, and that of their relatives, alleviating pain and other symptoms, as well as providing spiritual and psychological support from the moment of diagnosis to the end of life and during bereavement [1].

The World Health Organization [2], the European Palliative Care Association [3] and the PC strategy of the Spanish National Health System [4] recommend a shared model of palliative care that ensures care continuity. Therefore, this calls for the intervention of the various care resources: primary healthcare, hospital and centers specialized in PC (home- or hospital-based).

The care of patients with palliative needs, as well as the care of their relatives and/or caregivers, continues to pose important challenges. These include the scarce and unequal training of nursing professionals in PC at all levels of care [5,6,7]. Several studies confirm the need to address training deficits and to improve preparation to ensure high-quality care [8,9,10,11,12,13,14]. In the same vein, the European Association for Palliative Care (EAPC) [15] and the Spanish Association of Nursing in Palliative Care (Spanish acronym AECPAL) [16], recommend basic training in palliative care to be taught by all universities offering qualifications in nursing. Such basic training should impart the minimum knowledge required to provide effective care in the different clinical scenarios commonly found for these patients and their relatives at any level of care [17]. The AECPAL [16] defines five basic knowledge areas: principles of palliative care, communication skills, symptom management and specific care plans, coping with loss and death, and ethical and legal issues.

Besides this, care is based on acquired knowledge, which is reflected by derived actions or implementation. Pedagogy advocates the need to apply what has been learned to new contexts through other skills after a period of reflection and through a transfer process [18]. Moreover, from a bioethics perspective, a pedagogical teaching approach based on the transmission of knowledge and procedures is established, as well as a Socratic approach to change attitudes or behaviors [19].

In this vein, a study involving hospital nurses demonstrated that although half of them had basic training in PC, only about 15% actually applied this knowledge in practice (use of symptom assessment scales, application of non-pharmacological measures or patient participation in decision-making) [20].

The above would indicate that not only is it important to acquire knowledge, but also that this knowledge should reach patients and their relatives through practical application in clinical practice. Indeed, not only is it important to know “what nurses know”, but also “what they do or apply”.

The concern for palliative care-related issues has led to associated studies from various perspectives. Many instruments have been created and validated in Spanish to measure (jointly or separately) the knowledge, attitudes, skills and competencies on the subject, of different professionals (doctors, nurses and nursing assistants). Each of these professionals is considered within their specific discipline-related competencies, performance of their functions in the end-of-life care of dying patients and of their relatives and at different levels.

Some of the instruments attempt to approach the care provided by professionals to patients indirectly, assessing fear or competence on facing death, or attitudes towards patient care at the end of life. The Collet-Lester Scale and Bugen’s Coping with Death Scale measures the improvement in death [21,22,23].

Other instruments evaluating attitudes nursing trainees and professionals towards end-of-life patient care are Attitudes about End-of-Life Care Scale, the Death Attitude Profile-Revised (DAP-R), the Frommelt Attitude Toward Care of the Dying scale (FATCOD) or the Palliative Care Attitudes Scale (PCAS) [24,25,26,27,28,29].

Other scales focus on measuring the perceived self-efficacy competencies or self-reporting confidence and the perceived educational needs in PC, such as the Self-Efficacy in Palliative Care Scale [30,31] and the Self-reporting Confidence and Educational Needs in Hospice Care developed by Kwon [32]. 

Some scales have been created to measure attitudes and knowledge such as the Palliative Care Knowledge Test (PCKT) for general practitioners and nurses, the Palliative Care Attitude and Knowledge questionnaire (PCAK) for physicians and, more recently, the Palliative Approach for Nursing Assistants Questionnaire (PANA) that adds skills [33,34,35,36]. 

The Palliative Care Quiz for Nurses (PCQN) measures nurses’ knowledge in palliative care of three areas: philosophy and principles of PC, pain and symptoms control and the psychosocial aspects of care [37]. 

The Rotterdam MOVE2PC Scale assesses knowledge, attitudes, values and skills for the study of competencies and educational needs in PC of general nurses. This scale evaluates nurses’ opinions and subjective norms related to PC, potentially difficult situations in the final weeks or days of a patient’s life and knowledge [38].

However, the previous instruments aimed at nurses, evaluating knowledge, do so from the subjectivity of the respondent; according to their perception of how prepared they feel in a given situation. Furthermore, none of them evaluates the implementation of the knowledge they possess; therefore, the actual patient care provided is unknown. 

These instruments usually take into account the common competences of the nursing discipline, but they do not usually take into account the tasks performed at the different levels of care and that, therefore, determine the type of care provided. The instrument should differ, depending on whether it is aimed at hospital nurses or primary and home healthcare nurses; based on their scope of care; and the possibility of monitoring family grief.

A validated instrument to assess theoretical knowledge and its application to clinical practice by primary or home healthcare nurses would enable specific training to be guided according to the shortcomings detected. This could have an impact on the provision of more effective palliative care for patients.

The purpose of the study was to develop and test the psychometric properties of the Investigación Cuidados Enfermeros/Investigation into Nurses’ Care Understanding of End-of-Life (INCUE) questionnaire, an instrument developed to assess the basic training needs in palliative care of primary and home healthcare nurses.

## 2. Materials and Methods

The INCUE questionnaire was developed based on a three-phase mixed-method design (Figure 1), based on the classical test theory and factor analysis models [39,40]. In the first phase, qualitative and quantitative methods and techniques were combined to delimit the framework or theoretical basis and generate the items [41,42]. In the second phase, the validity of the content and the internal consistency of the instrument were assessed. In the third phase, the psychometric properties of the instrument were tested by confirmatory factor analysis (CFA), to evaluate validity and reliability. According to Batista-Foguet, Coenders and Alonso the confirmatory perspective assigns specific indicators to specific dimensions. In this way, CFA models allow validity to be compared, adjusting a model that assumes it and diagnosing its goodness-of-fit (construct validation). In this type of model, each item saturates only on the factor-dimension of which it should constitute a valid indicator [39].

### 2.1. Phase 1: Theoretical Basis and Generation of Items

In the first step, a literature review about nursing competencies and training in PC was performed, objectives were set, the construct was defined, the dimensions and indicators to be measured were determined, and items and their response types were listed. The selection criteria (advanced PC training and having a minimum of 10 years’clinical experience) and number of experts were established, the evaluation template was designed, and the concordance between judges was calculated following the criteria of Escobar-Pérez and Martínez [42].

After reviewing the literature and the existing instruments, the construct was defined based on the five basic training areas in PC for nurses described by AECPAL [16]: principles of palliative care (A1); symptom management and specific care plans (A2); coping with loss and death (A3); communication skills (A4), and ethical and legal issues (A5).

The first panel of experts evaluated the results of the literature review and the proposed objectives. Using the Delphi method, they generated the items based on the document outlining the recommendations of the AECPAL for Nursing Degree training [43], which specifies the basic theoretical and practical knowledge required for the end-of-life care of dying patients and their relatives. This knowledge was considered as the minimum benchmark to care for these patients and their relatives effectively in the different clinical scenarios commonly faced by the primary or home healthcare nurse in undertaking his/her work duties.

The evaluation of theoretical knowledge and its application to clinical practice were established as objectives of each of the areas. Agreement was reached on the number of items required for each area (Table 1).

Subsequently, the first panel of experts established the selection criteria for the second panel of experts and the item evaluation sheets to assess the suitability, clarity and relevance of the items, as well as additional contributions. The panel of experts should comprise at least 10 nurses with advanced PC training and have a minimum of 10 years’ clinical experience.

In the second step, the first version of the questionnaire was submitted to expert judgment by Cronbach’s alpha test and Kendall’s W test, to evaluate its reliability, internal consistency and expert consensus. Kendall’s W test also compares the potential association between ordinal variables in related samples and tests the null hypothesis of lack of agreement between judges. We performed four Kendall tests, one for each criterion and one for the three criteria together (adequacy, clarity and relevance). Additional textual contributions were collected for content analysis and applied to the pilot questionnaire [44].

### 2.2. Phase 2: Pilot Study

In this phase, descriptive analyses were performed to analyze item comprehensibility using a Likert scale from 1 to 5, scoring from lowest to highest comprehensibility.

Different parameters were evaluated to determine the consistency of the items within each area. The KR20 coefficient was calculated for the items in the theoretical block, whereas the Cronbach’s alpha was applied to the practical items. The values of these coefficients range between 0 and 1 and are considered acceptable when they are equal to or greater than 0.7 in order to ensure scale reliability [45].

### 2.3. Phase 3: Validation

In the validation phase, various analyses were performed in order to assess the reliability and validity of the questionnaire responses. In the theoretical block they were analyzed according to the right/wrong answers of the respondents. To evaluate reliability, internal consistency was analyzed, calculating KR20 and Cronbach’s alpha coefficients for the items in the theoretical and practical parts, respectively [45]. Finally, the homogeneity index or item-total correlation of each item was calculated, indicating the correlation between the score in each item and the sum of the scores in the remaining items. To evaluate the construct validity of the instrument, CFA was performed using psych software [46] and lavaan R package [47].

As a previous step to this analysis and to verify the CFA suitability, the KMO measure of sampling adequacy was calculated [48]. In this coefficient, a value of less than 0.5 indicates that the correlation between the variables is not sufficiently significant; therefore, it is inadvisable to analyze the inter-variable relationships. Furthermore, the Bartlett sphericity test was performed to verify that the correlations between the variables are not null [49].

The most commonly used indices to assess the fit of a CFA model are the Comparative Fit Index (CFI), the Tucker Lewis Index (TLI) and the Root Mean Squared Error Approximation (RMSEA). In order to consider that a model has a good fit, the following values are recommended: CFI ≥ 0.9, TLI ≥ 0.95 and RMSEA < 0.08 [50].

As in the pilot phase, the data were initially subjected to descriptive analysis to assess item comprehensibility.

Statistical software R (R Foundation for Statistical Computing, Vienna, Austria.) was used for data analysis (version 4.0.2).

### 2.4. Sample and Data Collection

The first panel of experts was composed of eight nurses with advanced PC training, extensive clinical and university teaching experience; with connections to the research group through AECPAL.

The second panel of experts was formed by 10 palliative-care nurses from AECPAL.

In the pilot phase, 31 nurses were recruited and included through non-probabilistic sampling during the months of November and December 2020. These participants shared similar characteristics to those who are targeted by the questionnaire under validation. All of them met the inclusion criteria as nurses working in the primary or home healthcare area in Spain.

Finally, for the validation phase there is no consensus regarding the size of the sample size needed to apply a factor analysis. Some authors set different criteria that varies between five and 10 participants per item or a minimum of 300 [51], while others only consider that a minimum of 300 is needed in order to obtain reliable results [52,53], or that at least five individuals per item are counted on [54,55].

In this phase, 344 nurses were recruited. Only 339 met the inclusion criteria. These were the same as in the pilot phase. The sampling was non-probabilistic during January and February 2021.

Given the exceptional circumstances created by the state of alarm in Spain, data collection was conducted online via Google Forms. The great added difficulty for the online selection of participants was to carry out probabilistic sampling due to the voluntary self-selection of the participants. To avoid this bias, we used recruitment strategies through natural leaders, nursing forums and Spanish associations for primary nursing by means of a snowball technique through social media where the target was nursing professionals, highlighting those participants who adequately represent all the strata of the population under study. These same circumstances have been evidenced in similar studies [56,57,58]. The online self-report questionnaire contained a brief introduction, the objective of the study, inclusion criteria and the need for consent for participation, guaranteed anonymity, confidentiality and the possibility of withdrawal.

### 2.5. Ethical Considerations

The present study was reviewed and approved by the Drug Research Ethics Committee of the Hospital Universitario Dr. Peset (Research Project EAPCP19-V01 and CEIM code 11/20). Potential study participants were provided with a detailed description of the study and were assured of confidentiality. Written, informed consent was obtained from each participant. They were also informed of the voluntary nature of the study participation and completion without any negative consequences.

## 3. Results

The participants’ socio-demographic data are shown in Table 2.

### 3.1. Phase 1: Theoretical Basis and Generation of Items

#### 3.1.1. Step 1

For the first version of the instrument, 67 items were drawn up (Table 1). Thirty-five items to assess theoretical knowledge (theoretical block), with dichotomous answers (yes, no, do not know—no answer). Thirty-two items evaluated the practical application or practical block (measured by a 5-point Likert scale: never, rarely, sometimes, frequently and always).

#### 3.1.2. Step 2

Kendall’s W test for inter-judge concordance resulted in 0.421 for relevance, 0.508 for clarity, 0.484 for adequacy and 0.429 for all three criteria together. In addition, a *p*-value very close to <0.001 was obtained in all four tests. On this basis, the null hypothesis (Ho) was rejected, concluding that concordance was assumed in the judges’ assessment.

In the experts’ judgment, the analyses showed a Cronbach’s alpha 0.960 in relevance, 0.978 in clarity and 0.967 in adequacy. However, it was decided to remove items with low scores on relevance and adequacy, thus eliminating one item from the practical block on relevance (A5) and two items from the theoretical block on adequacy (A1 and A4). Items with low clarity scores were reformulated as a result of textual contributions from experts, through consensus. This affected one item from the theoretical block and another from the practical block in clarity (both from A4).

These modifications led to the second version of the instrument, with 33 items in the theoretical block and 31 in the practical block.

### 3.2. Phase 2: Pilot Study

The average comprehension of the items of the theoretical block was 4.69 and 4.76 for the practical block, with a maximum score of 5.

However, those items that obtained a score below 3 in comprehensibility by one of the subjects were reformulated by expert consensus. Consequently, the items in theoretical block 5 and 6 (A1), 10 and 12 (A2), 20 (A3) and 25 (A4) and practical block 53 (A4) and 63 (A5) were modified.

In this phase the KR20 coefficient for the theoretical block was 0.828.

Cronbach’s alpha for the practical block was 0.943.

These values indicated internal consistency to both the theoretical and practical part, giving rise to the third version of the questionnaire.

### 3.3. Phase 3: Validation

The mean comprehension score was 4.68 in the theoretical block and 4.78 in the practical block.

An average of 17 min was taken to complete the questionnaire.

In the reliability analysis to determine the internal consistency of each part or block of the questionnaire (theoretical and practical) items 7, 14 and 16 of the theoretical block were deleted as they did not present variability (all subjects answered the same option).

Some items were observed to have very low homogeneity indices (below 0.1). Then, the items with the lowest item-total correlation were eliminated one by one until all these indices were equal to or greater than 0.1. Following this criterion, items 6 and 22 of the same theoretical block were discarded. In the practical block all items showed reliability.

The KMO measure of sampling adequacy in the set of variables was 0.873.

Bartlett’s sphericity test provided sufficient statistical evidence to disprove that the correlation matrix was an identity matrix; accordingly factor analysis was performed in both blocks (χ^2^ = 8386, df = 1830, *p*-value = <0.001).

The model was constructed without the previously discarded items (6, 7, 14, 16 and 22). When constructing this model, it was observed that the estimated variance of item 3 was negative and its weight in the model factor greater than 1 (Heywood case), therefore this item was removed from the analysis. Item 19 presented multicollinearity with items 47 and 55 in the practical block; therefore, it was removed from the model. Item 62 in the practical block led to convergence problems in the model, and therefore it was also discarded.

Finally, after deleting these items, the resulting model presented a good fit. Again, we assessed the reliability of the constructs through internal consistency analysis. In this case, item 27 presented a homogeneity index lower than 0.1. Therefore, the CFA was discarded and repeated. After this, item 11 presented convergence problems and was removed from the model.

Subsequently, the panel of experts decided to withdraw item 28 to unify the evaluation criteria by areas of the instrument, even though it fulfilled statistical criteria.

Questionnaire reliability was 0.700 in the theoretical block (KR20 Index) and 0.941 in the practical block (Cronbach’s alpha).

Table 3 shows the load or weight of the items in the factors of the model without the deleted items and the final internal consistency and correlation between the items of the theoretical and practical block.

After the modifications described above, the model converges and shows an adequate fit, specifically CFI = 0.977, TLI = 0.977 and RMSEA = 0.05. The correlation between the two factors in the model is ρ = 0.63.

## 4. Discussion

The psychometric testing of the INCUE questionnaire indicated that the model defined by the researchers is adequate and presents a good fit, according to the criteria established in the CFA [39] regarding reliability, internal consistency and validity of the construct. Both theoretical and practical blocks demonstrated separate internal consistency. This property would make independent use feasible. However, the CFA considers both blocks together.

The reliability scores (KR20 0.700 and α 0.941) are similar to those obtained by other instruments in Spanish such as the PCQN (KR20 0.72 and α 0.67) [36], Rotterdam MOVE2PC (α 0,786) [37], EACP (α 0.807) [29], or the Self-Efficacy in Palliative Care Scale (α 0.944) [31], or that of other instruments developed and validated in English such as the PCKT (α 0.87) [33] or the three PANA questionnaires (α 0.69, α 0.81 and α 0.80) [38].

The INCUE questionnaire differs from the other instruments cited herein in that it evaluates knowledge and practical application of the same, giving similar importance to all areas of care [21,22,36]. What is more, the INCUE questionnaire evaluates the practical application of knowledge in the care of people with palliative needs, and the frequency (never, rarely, sometimes, frequently and always) with which this is performed. This is not assessed by any of the other instruments reviewed. Subjectivity of the respondent is avoided by eliminating questions as to whether he/she feels prepared [31] or placing them in a hypothetical care-related situation [38]. In this respect, a nurse may feel prepared to perform the care-related task but, nevertheless, classify the situation as difficult. The crux of the matter is whether he/she actually put this into practice, despite the complexity.

Furthermore, the INCUE questionnaire enables a correlation to be drawn between the theoretical block (knowledge or what we know) and its application in the practice of care (what we do), since this correlation is not always proportional [20]. Establishing in which areas primary or home healthcare nurses have shortcomings in knowledge or in its practical application would facilitate the identification of training needs and subsequent training pinpointing these areas. In this way, nurses could acquire the minimum or basic knowledge necessary for end-of-life care of patients and their relatives. In addition, shortcomings in practical application despite sufficient theoretical knowledge could be due to factors other than training (overload, lack of material resources, etc.) and would thus require assessment to rule out these potential causes.

Also, one desirable quality in an instrument is discrimination of participants. In this respect, the INCUE questionnaire differs from other instruments in that it exclusively targets nurses, adapting to the competencies of this discipline and the care they perform [33]. In the same vein, it is not aimed at nursing students. We understand that they are still in the process of acquiring knowledge and their clinical practice is not independent, but supervised.

In addition, the questionnaire targets only primary and home healthcare nurses in the Spanish context. Although the theoretical block is applicable to all nurses, regardless of where they practice their profession, the practical block is not applicable to hospital nurses. Only home healthcare nurses can monitor patients and those grieving at home and respond to items 32, 38, 39, 40 and 41 (Appendix A).

Different scoring criteria were established for the theoretical and practical block, to reach the minimum standard necessary to effectively care for these patients and their relatives. In the first block, agreement was reached to obtain 75% of correct answers in the areas of four items (1 and 4) and 80% in the areas of five items (2, 3 and 5), allowing only one error per area. For the practical block, a minimum score of 18 points per area was established, out of a total of 24 potential points; also representing 75% of the total. The score given to each response was 0 points to “never“, 1 point to “rarely”, 2 points if they respond “sometimes”, 3 points to “frequently”, and 4 points to “always”. Any area in which the threshold score is not obtained will require training.

The final questionnaire consists of 23 items in the theoretical block and 30 in the practical block, and is effective and user-friendly due to its duration.

A few limitations should be taken into consideration in interpreting the results of this study. First, the validation procedure has been described in detail to provide full process information. The use of a panel of experts is criticized by some authors due to possible alterations in validity [59]. However, recommendations were followed with respect to the number of experts [60] on the double panel and the necessary methodological and statistical rigor measures were taken to interpret and apply their results correctly [42].

During the online selection of the sample the participants’ IP was not tracked, given that it does not guarantee the hindrance of a subject answering through two different IP directions. Different individuals can use the same IP in a professional context nonetheless.

Although the sample size in the psychometric tests was lower than that of some studies undertaken to develop and validate instruments [33], it was similar to others [17,36]. Moreover, it exceeded the number recommended by some authors [51,53,54,55], and that of other similar studies [30,35]; being sufficient for statistical analysis. Second, the questionnaire has been validated in the Spanish context and targets primary and home healthcare nurses, thus it is not applicable in other contexts or professional profiles. It would be interesting to validate the INCUE questionnaire in other care-related settings or contexts, once suitably adapted.

## 5. Conclusions

The INCUE questionnaire is suitable to assess knowledge in palliative care and its application in clinical practice of primary and/or community healthcare nurses and in the home environment. This instrument pinpoints specific shortcomings in the defined areas, identifying training needs. This facilitates specific training based on these needs, in order to acquire the minimum or basic knowledge necessary for the end-of-life care of dying patients and of their relatives.

Those responsible for the training of nurses need to objectively know the gaps and training needs when planning an effective training program. Comprehensive end-of-life care for patients and their relatives requires nurses to acquire and to exercise their skills.

## Figures and Tables

**Figure 1 ijerph-18-10995-f001:**
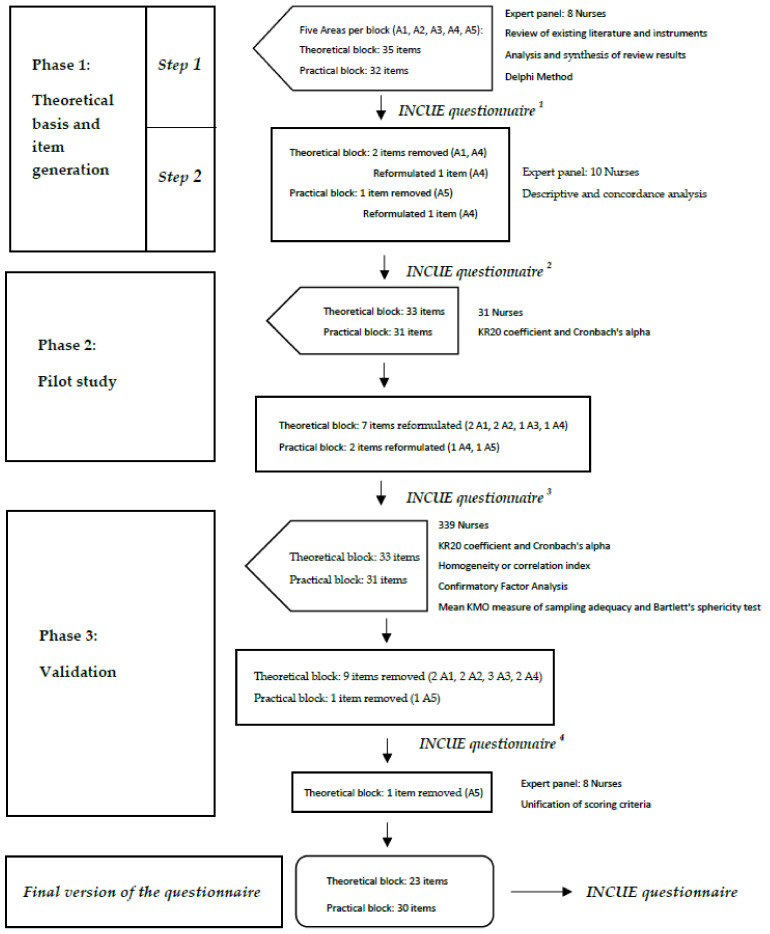
Investigación Cuidados Enfermeros/Investigation into Nurses’ Care Understanding of End-of-Life (INCUE) questionnaire developmental process.

**Table 1 ijerph-18-10995-t001:** The INCUE questionnaire one item generation.

Areas	Part 1: Theoretical Knowledge Theoretical Block (*n* = 35)		Part 2: Practical Application Practical Block (*n* = 32)	
	Number of Items	Response Type	Number of Items	Response Type
1. Principles of PC	7	Yes	6	Never
2. Symptom management and specific care plans	7		6	Rarely
3. Coping with loss and death	8	No	6	Sometimes
4. Communication skills	7		6	Often
5. Ethical and legal aspects	6	Do not know/no answer	8	Always

Abbreviations: Palliative Care (PC).

**Table 2 ijerph-18-10995-t002:** Sample socio-demographic data.

Variable	Expert Panel 1 (*n* = 8)	Expert Panel 2 (*n* = 10)	Pilot Phase (*n* = 31)	Validation Phase (*n* = 339)
Age (years), M ± SD	49.6 ± 1.3	45.7 ± 12	41.8 ± 12	45.5 ± 11.2
Gender, *n* (%)				
Female	7 (75)	9 (90)	27 (87.1)	280 (82.6)
Male	1 (25)	1 (10)	4 (12.9)	59 (17.4)
Maximum level of professional qualification				
Doctorate	2 (25)	2 (20)	1 (3.2)	17 (5)
Nursing Specialization	0 (0)	0 (0)	2 (6.5)	53 (15.6)
Master degree	6 (75)	8 (80)	5 (16.1)	88 (26)
Diploma/Graduate	0 (0)	0 (0)	23 (74.2)	181 (53.4)
Current position				
Reference Nurse in Palliative Care	0 (0)	10 (100)	1 (3.2)	45 (13.3)
Nurse and Center Coordinator	0 (0)	0 (0)	2 (6.5)	0 (0)
Nurse	8 (100)	0 (0)	27 (87.1)	208 (61.4)
Center Coordinator	0 (0)	0 (0)	1 (3.2)	33 (9.7)
Community case manager	0 (0)	0 (0)	0 (0)	53 (15.6)
Professional experience (years), M ± SD	29.2 ± 2.2	20.7 ± 6.5	13.7 ± 9.18	21 ± 11.7
Palliative-care training				
Yes	8 (100)	10 (100)	25 (80.6)	292 (86.1)
No	0 (0)	0 (0)	6 (19.4)	47 (13.9)
Level of training in PC				
Advanced (Master or PhD)	8 (100)	10 (100)	0 (0)	51 (15.0)
Intermediate (80–150 h)	0 (0)	0 (0)	10 (32.2)	92 (27.2)
Basic (25–80 h)	0 (0)	0 (0)	15 (48.4)	154 (45.4)
Don’t know/No answer	0 (0)	0 (0)	6 (19.4)	42 (12.4)

Abbreviations: Mean (M), Standard deviation (SD), hours (h).

**Table 3 ijerph-18-10995-t003:** Factors weighting of the items in the model and final block reliability measures.

Theoretical Block (without the Questions 3, 6, 7, 11, 14, 16, 19, 22, 27 and 28)	Practical Block (without Question 62)
Item	Factor Weight	Item-Total Correlation	KR20 on Omitting the Question	KR20 Index	Item	Factor Weight	Item-Total Correlation	Cronbach’s Alphaon Omitting the Question	Cronbach’s Alpha
1	0.56	0.109	0.700	0.700	34	0.69	0.415	0.941	0.941
2	0.64	0.242	0.693		35	0.76	0.758	0.938	
4	0.43	0.263	0.690		36	0.80	0.706	0.939	
5	0.52	0.255	0.692		37	0.71	0.635	0.939	
8	0.75	0.432	0.677		38	0.69	0.650	0.939	
9	0.37	0.206	0.695		39	0.68	0.547	0.940	
10	0.24	0.119	0.705		40	0.74	0.689	0.938	
12	0.55	0.348	0.681		41	0.61	0.653	0.939	
13	0.57	0.290	0.688		42	0.60	0.695	0.938	
15	0.75	0.171	0.698		43	0.49	0.395	0.942	
17	0.44	0.304	0.686		44	0.81	0.681	0.938	
18	0.68	0.339	0.686		45	0.65	0.701	0.938	
20	0.39	0.148	0.703		46	0.83	0.776	0.937	
21	0.77	0.357	0.680		47	0.84	0.660	0.939	
23	0.46	0.162	0.697		48	0.66	0.649	0.939	
24	0.2	0.212	0.697		49	0.73	0.658	0.939	
25	0.54	0.272	0.689		50	0.64	0.672	0.938	
26	0.48	0.384	0.678		51	0.52	0.579	0.940	
29	0.62	0.213	0.695		52	0.36	0.568	0.940	
30	0.46	0.223	0.694		53	0.68	0.460	0.942	
31	0.45	0.218	0.694		54	0.69	0.680	0.939	
32	0.47	0.339	0.683		55	0.76	0.577	0.940	
33	0.45	0.351	0.681		56	0.80	0.740	0.938	
					57	0.71	0.782	0.938	
					58	0.69	0.557	0.940	
					59	0.68	0.663	0.939	
					60	0.74	0.629	0.939	
					61	0.61	0.429	0.941	
					63	0.60	0.356	0.942	
					64	0.49	0.605	0.940

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
