# Peer review of "Design and Validation of the INCUE Questionnaire: Assessment of Primary Healthcare Nurses’ Basic Training Needs in Palliative Care"

_ijerph, 2021, doi:10.3390/ijerph182010995_

Round 1
Reviewer 1 Report
Dear Authors
Thank you for this interesting manuscript. I do have some minor clarifications I would like you to respond to.

Author Response
International Journal of Environmental Research and Public Health.
October 12 ,2021
Dear Reviewer 1:
We appreciate the journal's interest in our article and the remarks of the reviewers, which have been very enriching.
We have included a detailed list of the changes that have been made, with our responses to the reviewers´ comments.
Phase 1: Theoretical basis and generation of items
- How a literature review was performed?
- How the selection criteria and number of experts were established?
Sample and Data Collection
- Teaching experience?
- Connections to the research group?
Results. Step 2
- How low clarity scores were reformulated?
Results. Phase 2: Pilot study
- How items that obtained a score below 3 in comprehensibility by one of the subjects were reformulated?
Answer: It has been included in the manuscript.
Results. Fase 3 Validación
- How did reason various analyses?
Answer: They are described below in the manuscript’s paragraph.
- Heywood case
Answer: A Heywood case occurs in factor analysis when the iterative maximum likelihood estimation method converges to unique (specific) variances values that are less than a prefixed lower bound value. Minitab sets the value for these unique variances equal to 0 and their corresponding communalities equal to 1. Heywood cases occur frequently when too many factors are extracted or the sample size is too small.
Sample and Data Collection
- The first panel of experts was composed of eight nurses of eight nurses. Educational level?
Answer: Educational level or maximum level of professional qualification are included in Table 2.
- In this phase, 344 nurses were recruited. Only 339 met the inclusion criteria. Short comment about the criterias for redability.
Answer: Sorry, but we do not understand the question. If you’re refering to the inclusion criteria, they are in the manuscript “These were the same as in the pilot phase”.
- Safety online via Google Forms?
Answer: Security and other measures are included in the limitations.
We hope that our answers are satisfactory. For that reason, we are resending the manuscript.
Should you have any doubts, do not hesitate on contacting us again.
Sincerely,
Encarna Chisbert-Alapont
Reviewer 2 Report
The research is very interesting.
This study aimed to develop and psychometrically test the INCUE questionnaire (INvestigación CUidados Enfermeros/ Investigation into Nurses’ Care Understanding of End-of-life) to assess basic training needs of primary or home healthcare nurses in palliative care.
The methodology was described in detail and the results were presented in detail and legibly.
A few limitations have taken into consideration in interpreting the results of this study.
I recommend the paper in present form for publication.
Author Response
Thank you very much for your time and your remarks.
Reviewer 3 Report
Authors sought to assess and develop the INCUE questionnaire and evaluate its usability towards assessing the basic training needs of nurses working within the palliative care healthcare settings. Using a series of specialist evaluated sequential steps and metrics, they reported the completion and validation of their designed questionnaire. The manuscript is well written.
The only minor concern as alluded by authors in Table 2 is the average years of professional experience of the nurses utilized in the study design process. It appears the number of young, low experienced nurses used as study subjects were few in number. It is understood that authors adopted non-probabilistic sampling method.
Author Response
International Journal of Environmental Research and Public Health.
October 12 ,2021
Dear Reviewer 3:
We appreciate the journal's interest in our article and the remarks of the reviewers, which have been very enriching.
We have included a detailed list of the changes that have been made, with our responses to the reviewers´ comments.
The method for sampling was non-probabilistic.
The average years of professional experience of the nurses that have participated in the study is high. We are not surprised given that in Spain (2012) 44,5% of primary care nurses are over 49 years old.
Reference: “Informe sobre profesionales de enfermería. Oferta-Necesidad 2010-2025” Available online: https://www.mscbs.gob.es/profesionales/formacion/necesidadEspecialistas/doc/21-NecesidadesEnfermeras(2010-2025).pdf
We hope that our answers are satisfactory. For that reason, we are resending the manuscript.
Should you have any doubts, do not hesitate on contacting us again.
Sincerely,
Encarna Chisbert-Alapont
Reviewer 4 Report
After article valuation "Design and validation of the INCUE questionnaire: Assessment of Primary Healthcare Nurses' Basic Training Needs in Palliative Care":
Research with great interest and practical applicability. The manuscript is of high quality. Congratulations to the authors
The introduction is adequate. It exposes the main concepts and analyzes the existing questionnaires adequately justifying their study objective. Correct theoretical foundation.
There is detailed description of the research methods used. The methods used are well supported by bibliography. It would be necessary to add bibliography on the Delphi method.
The results shown are concrete and detailed, explaining how to obtain this information and what scientific evidence it has.
Discussion and conclusions are appropriate.
Author Response
International Journal of Environmental Research and Public Health.
October 12 ,2021
Dear Reviewer 4:
We appreciate the journal's interest in our article and the remarks of the reviewers, which have been very enriching.
We have included a detailed list of the changes that have been made, with our responses to the reviewers´ comments.
In the consensus of experts, the Delphi method is usually used. We have not seen it referenced in other articles about the validation of instruments reviewed. For this reason, we do not reference it.
We hope that our answers are satisfactory. For that reason, we are resending the manuscript.
Should you have any doubts, do not hesitate on contacting us again.
Sincerely,
Encarna Chisbert-Alapont